# Rational Use of CT-Scan for the Diagnosis of Pneumonia: Comparative Accuracy of Different Strategies

**DOI:** 10.3390/jcm8040514

**Published:** 2019-04-15

**Authors:** Nicolas Garin, Christophe Marti, Sebastian Carballo, Pauline Darbellay Farhoumand, Xavier Montet, Xavier Roux, Max Scheffler, Christine Serratrice, Jacques Serratrice, Yann-Erick Claessens, Xavier Duval, Paul Loubet, Jérôme Stirnemann, Virginie Prendki

**Affiliations:** 1Department of Internal Medicine, Riviera-Chablais Hospitals, 1870 Monthey, Switzerland; 2Department of Internal Medicine, Geneva University Hospitals, 1205 Geneva, Switzerland; Christophe.Marti@hcuge.ch (C.M.); Sebastian.Carballo@hcuge.ch (S.C.); Pauline.Darbellay@hcuge.ch (P.D.F.); Jacques.Serratrice@hcuge.ch (J.S.); Jerome.Stirnemann@hcuge.ch (J.S.); 3Faculty of Medicine, University of Geneva, 1211 Geneva, Switzerland; Xavier.Montet@hcuge.ch (X.M.); Virginie.Prendki@hcuge.ch (V.P.); 4Department of Radiology, Geneva University Hospitals, 1205 Geneva, Switzerland; Max.Scheffler@hcuge.ch; 5Department of Rehabilitation and Geriatrics, Geneva University Hospitals, 1205 Geneva, Switzerland; Xavier.Roux@hcuge.ch (X.R.); Christine.Serratrice@hcuge.ch (C.S.); 6Department of Anesthesiology, Pharmacology and Intensive Care, Geneva University Hospitals, 1205 Geneva, Switzerland; 7Department of Emergency Medicine, Centre Hospitalier Princesse Grace, 98000 Monaco, Monaco; ye.claessens@gmail.com; 8Department of Infectious Disease, Bichat-Claude Bernard University Hospital, 75877 Paris, France; Xavier.Duval@aphp.fr (X.D.); Paul.Loubet@aphp.fr (P.L.); 9INSERM, IAME, UMR 1137, 75870 Paris, France

**Keywords:** clinical prediction model, lower respiratory tract infection, CT-scan

## Abstract

Diagnosing pneumonia in emergency departments is challenging because the accuracy of symptoms, signs and laboratory tests is limited. As a confirmation test, chest X-ray has significant limitations and is outperformed by CT-scan. However, obtaining a CT-scan in all cases of suspected pneumonia has significant drawbacks. We used a cohort of 200 consecutive elderly patients admitted to the hospital for suspected pneumonia to build a simple prediction score, which was used to determine indication for performing a CT-scan. The reference diagnosis was adjudicated by experts considering all available data, including evolution until discharge and CT scan in all patients. Results were externally validated in a second cohort of 319 patients. Pneumonia was confirmed in 133 patients (67%). Area under the receiver operator curve (AUROC) of physician evaluation was 0.55 (0.46–0.64). The score incorporated four variables independently predicting confirmed pneumonia: male gender, acute cough, C-reactive protein >70 mg/L, and urea <7 mmol/L. AUROC of the score was 0.68 (95% confidence interval (CI) 0.60–0.76). When a CT-scan was obtained for patients at low or intermediate predicted risk (108 patients, 54% of the cohort), AUROC was 0.71 (0.63–0.80) and 0.69 (0.64–0.74) in the derivation and validation cohort, respectively. A simple prediction score for pneumonia had moderate accuracy and could guide the performance of a CT-scan.

## 1. Introduction

Accurately diagnosing pneumonia is a major challenge in emergency departments and ambulatory settings. The current reference diagnosis is the presence of an acute infiltrate on chest X-ray (CXR) along with consistent symptoms and signs [1]. However, symptoms and signs of pneumonia are neither sensitive nor specific, particularly in the elderly [2,3]. As a confirmation test, CXR lacks both sensitivity and specificity when compared with computed tomography scan (CT-scan) [4,5], and interpersonal agreement in the interpretation of CXR is low [6]. In elderly patients, the high incidence of other common causes of respiratory symptoms and CXR alterations (e.g., heart failure, acute exacerbation of chronic obstructive pulmonary disease, or cancer) further jeopardizes the accuracy of CXR [7].

Biomarkers (e.g., C-reactive protein or procalcitonin) only modestly improve the accuracy of clinical diagnosis [8,9,10].

Clinical prediction models have been developed to assess the probability of the disease but have limited discrimination [11]. Further, prediction models have been developed mostly in ambulatory settings, enrolling a population of relatively young patients with few comorbidities and a low prevalence of pneumonia. Finally, all prediction models have been tested against a reference diagnosis based on CXR, entailing a substantial risk of misclassification [5].

In a cohort of elderly patients admitted to the hospital for suspected pneumonia, low-dose computed tomography scan (LDCT) modified the probability of pneumonia in 45% of patients, with a net reclassification index of 8% [12]. However, obtaining LDCT in all patients suspected of pneumonia would be resource-demanding, expose some patients to unwarranted irradiation, and may lead to numerous incidental findings, whose potential negative impacts are not well appreciated [13]. Hence, identification of patients whose diagnosis and management is likely to be modified by LDCT would be welcome.

We aimed to build a simple prediction model allowing to determine indication for performing a CT-scan, and compared the accuracy of two strategies for the diagnosis of pneumonia in elderly patients in the emergency department: one based on the prediction model with LDCT performance in patients with predicted low or intermediate probability of disease, and the other based on LDCT performance in all patients (Figure 1). The reference diagnosis was based on CT-scan imaging and consensus expert evaluation.

## 2. Experimental Section

### 2.1. Setting and Patients

The setting and population have been described in detail elsewhere [12]. Briefly, we included consecutive patients admitted from the emergency department with suspected pneumonia after an evaluation based on standard practice (clinical evaluation, routine blood tests including C-reactive protein, and CXR). Patients were older than 65 years and had to present signs and symptoms suggestive of a respiratory infection severe enough to warrant antibiotic treatment according to admitting physicians. Performance of a CXR was mandatory. Presence of an acute infiltrate was assessed by the physician in charge of the patient. A LDCT without the injection of a contrast medium was performed in all patients within 24 h after inclusion and was interpreted by the radiologist on duty. Before and after the performance of the LDCT, physicians in charge were asked to assess the likelihood of pneumonia on a three-level Likert scale (high, intermediate, or low level of certainty) according to their clinical evaluation, incorporating results of blood tests and radiologic studies (CXR initially, then LDCT). We merged high and intermediate levels of certainty in one category (presence of pneumonia).

### 2.2. Diagnostic Criteria

The reference standard was adjudicated by a panel of three experts in the field supported by a radiologist expert in chest imaging, who assessed the diagnosis of pneumonia using all available information, including interpretation and images of the LDCT and evolution until hospital discharge or death. The experts rated the presence of pneumonia on a three-level Likert scale. When the diagnosis of pneumonia was dismissed, the experts assessed the most likely alternative diagnosis. High or intermediate probabilities were merged to form a binary variable (presence or absence of pneumonia).

### 2.3. Analysis

We explored the univariate association of data obtained during standard evaluation of patients with the presence of pneumonia according to the reference standard. Variables included age, gender, symptoms, physical signs, blood tests, and presence of a pulmonary infiltrate on CXR according to the interpretation of the physician in charge. All variables with a p value <0.20 were then entered in a multivariate logistic regression model with backward automatic selection, with confirmed pneumonia as the dependent variable. Finally, we built a score predicting the presence of pneumonia using variables independently associated with the presence of the disease. We determined optimal cut-offs for continuous variables by observation of the Receiver Operating Curves (ROC), and used unweighted coefficients. We then compared the accuracy of physician diagnosis before and after the performance of the LDCT with a diagnostic algorithm based on the score and mandating the performance of LDCT for patients with a low or intermediate predicted probability of pneumonia.

As external validation, we evaluated the accuracy of the algorithm in a second cohort composed of 319 adult patients visiting the emergency department of four French tertiary hospitals for suspected pneumonia who all had a CT-scan performed [5]. The reference diagnosis was adjudicated in a similar way as in the derivation cohort (expert assessment at 28 days using all available information, including CT-scan images and interpretation).

We used frequencies, percentage, means with standard deviations, and medians with interquartile range for descriptive statistics. We used one-way analysis of variance (ANOVA), Fisher’s exact test or Chi-square test as appropriate for univariate comparisons, and logistic regression with automatic conditional backward analysis for multivariate comparisons. Agreement between the physician’s assessment (with and without LDCT) and the reference diagnosis was evaluated with Kappa coefficient. We computed area under Receiver Operating Curves (AUROC), sensitivity, specificity, positive and negative predictive values, positive and negative likelihood ratios, and diagnostic odd ratio (DOR) to assess the accuracy of the different diagnostic strategies.

All confidence intervals are two-sided and a *p* value < 0.05 was considered significant. The Geneva institutional review board approved the study protocol (CER 14-250), which was registered in clinicaltrials.gov (NCT02467192).

## 3. Results

Two-hundred patients (median age 84 years, 51% males, 14% living in a nursing home) were available for the analysis. Forty-seven patients (24%) had a history of chronic respiratory disease, mostly chronic obstructive pulmonary disease, and 16 (8%) had bronchiectasis. The probability of pneumonia was high or intermediate in 183 patients (92%) according to the physicians in charge. A final diagnosis of pneumonia was confirmed by the experts in 133 (67%). Main alternative diagnoses were non-respiratory infections, bronchitis, heart failure, and exacerbation of chronic obstructive pulmonary disease. AUROC of physician evaluation without LDCT was 0.55 (0.46–0.64).

Univariate association of patients’ characteristics with the final diagnosis of pneumonia are displayed in Table 1.

In the multivariate model, four characteristics were independently associated with the diagnosis: male gender, acute cough, C-reactive protein, and urea (Table 2). The model was robust when incorporating creatinine instead of urea. The presence of a pulmonary infiltrate on CXR as assessed by the physician was not independently associated with the diagnosis.

The score predicting confirmed pneumonia assigned one point for the presence of acute cough, male gender, a C-reactive protein >70 mg /L, and urea <7 mmol/L. The prevalence of pneumonia increased with the number of points (Table 3). The AUROC of the score was 0.68 (95% CI 0.60–0.76). The algorithm mandated the performance of a LDCT in 108 patients with less than three points (54% of the population).

The validation cohort consisted of 319 patients (mean age 64.7 years, 49% males, 4% living in a nursing home). Chronic respiratory disease was present in 89 patients (28%), and bronchiectasis was identified by LDCT in 52 patients (16%). Pneumonia was confirmed in 156 (49%) patients. The prevalence of pneumonia ranged from 29% in patients with zero points to 93% in patients with four points. AUROC was 0.61 (95% CI 0.55–0.67). Prevalence of pneumonia was 45% in patients with less than three points and 60% in patients with three or four points.

The accuracy of the different assessments of the presence of pneumonia (physician in charge with and without LDCT; and the algorithm in the derivation and validation cohorts) is presented in Table 4. The agreement with the reference diagnosis for physician assessment before LDCT was only slight (kappa coefficient 0.12). After LDCT, physician assessment had substantial agreement with the reference diagnosis (kappa coefficient 0.63). The higher AUROC (0.80) and DOR (29) were for physician assessment with LDCT, the lower for physician assessment without LDCT (AUROC 0.55, DOR 3). In an analysis stratified on the score, sensitivity and specificity were 93% and 15% without LDCT in patients with zero to two points, and 88% and 67% after LDCT. Sensitivity and specificity in patients with more than two points were 96% and 16% before LDCT, and 95% and 74% after LDCT. The algorithm had intermediate accuracy (AUROC 0.71, DOR 18). In the validation cohort, the algorithm had similar accuracy, with an AUROC of 0.69 and DOR of 8.

## 4. Discussion

Our main finding is that using a simple prediction score in patients suspected of pneumonia allowed to forego performing a LDCT in nearly half the patients, with moderate accuracy.

Four characteristics of the patients were independently related with a diagnosis of pneumonia. Male gender has been consistently found in the literature as a risk factor for the disease [14,15]. This may be related to hormonal status, or more probably to a higher burden of comorbidities in men. Acute cough is part of some, but not all, clinical prediction models [11]. C-reactive protein is also consistently associated with the presence of pneumonia [9]. The best cut-off in our study was higher than in previous works [8], which can be explained by differences in setting (emergency department vs. ambulatory setting), population included (older vs. younger patients), and severity of disease.

The inverse relationship between urea (and creatinine) and presence of pneumonia, with lower values associated with a higher risk of the disease, was unexpected. Elevated urea is a well-known marker of severity of pneumonia, but has not been described elsewhere as associated with the presence of the disease. Chronic kidney disease is a risk factor for the development of pneumonia, but the relation should have been opposite to the one found in our study [16]. An inverse relation could stem from a more difficult radiologic diagnosis of pneumonia in dehydrated and hypovolemic patients, a frequently claimed interaction, though data supporting this issue are scarce and contradictory [17,18]. Finally, in this population of very old patients, where complaints and physical findings are poorly specific, dehydration or renal failure may be more strongly associated with alternative diagnoses to pneumonia, like cardiorenal syndrome or non-respiratory sepsis, explaining our finding.

Physician identification of a pulmonary infiltrate on CXR was not associated independently with the disease. Although disappointing, this finding is not unexpected. Previous studies have pointed to the low sensitivity and specificity of CXR for pneumonia, translating into the low confidence of clinicians in CXR for the diagnosis of pneumonia [19]. Major discordance between emergency physician and expert radiologist diagnosis have been described [20]. To build a score based on information readily obtained in an emergency setting, we did not include the radiologist interpretation of CXR in our score, as it is not widely available on a real-time basis.

Although the prevalence of pneumonia increased with the number of points, the accuracy of the score was rather low (AUROC 0.68). The algorithm mandating performance of a LDCT in patients with a score of zero to two resulted in an AUROC of 0.71 and a DOR of 18, with a LDCT performed in 54 % of the patients.

In a large study investigating the accuracy of a clinical prediction model for the presence of pneumonia, AUROC was 0.70 without, and 0.77 with the addition of C-reactive protein [8]. However, patients were recruited in the ambulatory setting (mean age 50 years), and the reference diagnosis was CXR, limiting the comparability of our findings. In a meta-analysis of individual patient data, AUROC of six clinical prediction models for the diagnosis of pneumonia in a primary care setting ranged from 0.53 to 0.79, with only two models having an area superior to 0.70 [11]. All had a reference diagnosis based on CXR. Hence, our score has comparable accuracy and is evaluated against a more robust reference diagnosis. Being based on only four variables without weighting, it is also easy to use in a busy emergency department.

Of note, the accuracy of the diagnosis assessed by the physician’s evaluation supplemented with LDCT in all patients was far from ideal: AUROC was only 0.80, with good sensitivity (92%) but only moderate specificity (69%). This point illustrates the difficulty of differentiating pneumonia from the numerous alternative diagnoses in elderly patients, even with advanced imaging. Pneumonia can be hard to differentiate from atelectasis, cardiogenic oedema, lung cancer, or lung infarct even on a LDCT. Moreover, more than one cause of lung opacification can coexist in the same patient.

Our study has some limitations. Principally, the derivation cohort had a small sample size. However, published studies enrolling consecutive patients with both a complete standard assessment and performance of a CT scan are scarce. Patients included were a particular population of very elderly patients with a high prevalence of pneumonia, and generalizability of our findings to other settings is not necessarily warranted. The association of renal markers (urea and creatinine) with pneumonia should be tested in other cohorts, as it has never been described before and might be specific to our cohort. The AUROC was similar in the validation cohort where included patients were significantly younger, but sensitivity and specificity differed markedly. Finally, global measures of accuracy like AUROC or DOR assign the same weight to false positive and false negative cases, although clinical implications differ.

The strength of our work is the use of a prospectively defined evaluation of the probability of disease by the physician in charge at different time points during the diagnosis process; the prospective design also allowed the collection and testing of a number of putative predictive variables without missing data. The reference diagnosis, based on a rigorous evaluation by experts and including LDCT in all patients, is less prone to misclassifications than the reference used in previous studies, based on CXR confirmation. We validated our findings in another cohort that used similar methods but included a quite different population.

Finally, before implementing the proposed score and algorithm in routine use, future studies should focus on the validation of its accuracy in larger populations from different settings. As for any diagnostic test, an implementation study should be conducted to test its effectiveness on clinical outcomes compared with the standard diagnostic pathway and to establish its cost-utility.

## 5. Conclusions

A clinical prediction score based on four easily available variables allows for more superior accuracy for the diagnosis of pneumonia than standard assessment. When targeting patients at low or intermediate predicted risk, LDCT is indicated for 54% of the patients. When a LDCT is obtained for all patients, the accuracy of the clinical diagnosis of pneumonia is only moderate when comparing with a reference diagnosis.

## Figures and Tables

**Figure 1 jcm-08-00514-f001:**
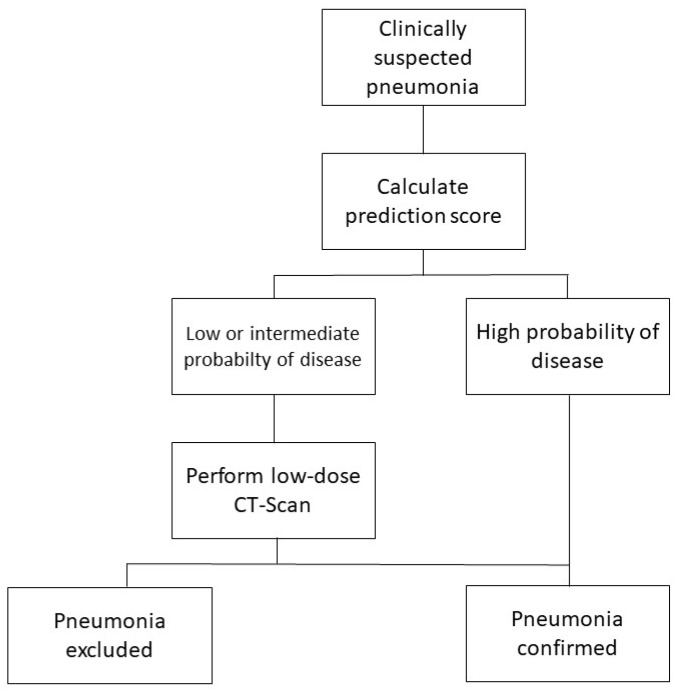
Algorithm to guide low-dose computed tomography scan (LDCT) performance.

**Table 1 jcm-08-00514-t001:** Univariate association between demographic, clinical, biological and radiological variables, and final diagnosis of pneumonia.

Variable	Pneumonia (N = 133)	No Pneumonia (N = 67)	*p-*Value
**Demographic Characteristics**			
Age	83 (78–89)	86 (80–92)	0.03
Male gender	71 (53)	28 (42)	0.18
Ambulatory setting (vs. nursing home or other)	117 (88)	55 (82)	0.28
**Symptoms**			
Acute cough	120 (90)	50 (75)	<0.01
Rales	114 (86)	57 (85)	1
Sputum production	49 (37)	25 (37)	1
Dyspnea	95 (71)	50 (75)	0.74
Chest pain	26 (20)	9 (15)	0.33
Confusion	60 (45)	32 (48)	0.6
**Signs**			
Heart rate	90 (78–104)	89 (77–101)	0.53
Respiratory rate	24 (20–28)	22 (15–24)	0.08
Temperature (°C)	38.0 (37.4–38.7)	37.7 (37.1–38.4)	0.07
Systolic blood pressure	131 (113–148)	138 (116–156)	0.2
Diastolic blood pressure	72 (62–83)	73 (67–85)	0.24
Hypoxemia (PaO2 < 8 kPa or SaO2 < 90%)	72 (54)	30 (45)	0.22
**Ancillary Tests**			
C-reactive protein (mg/L)	101 (59–135)	63 (38–108)	<0.01
Leucocytes (G/L)	12.0 (5.6)	10.7 (4.0)	0.09
Procalcitonin µg/L	0.36 (0.14–1.93)	0.25 (0.11–0.66)	0.04
Urea (mmol/L)	7.7 (5.7–10.8)	8.3 (6.2–12.9)	0.04
Creatinine (µg/L)	92 (69–125)	101 (83–141)	0.03
Probability of pneumonia on CXR (according to physician)			<0.01
Low	31 (23)	26 (39)	
Intermediate	36 (27)	23 (34)	
High	66 (50)	18 (27)	

Data are frequencies with percentage, and median with interquartile range. CXR: chest X-ray.

**Table 2 jcm-08-00514-t002:** Association of predictor variables with the presence of pneumonia.

Variable	Odd Ratio (95% CI)	*p*-Value
Male gender	2.23 (1.12–4.44)	0.022
Acute cough	3.77 (1.51–9.40)	0.004
C-reactive protein (mg/dL)	1.01 (1.00–1.01) ^1^	<0.001
Urea (mmol/L)	0.92 (0.86–0.98) ^2^	0.007

^1^ per mg/dL increment; ^2^ per mmol/L increment. CI: confidence interval.

**Table 3 jcm-08-00514-t003:** Number of patients with suspected pneumonia and prevalence of confirmed pneumonia according to the score.

Number of Points	Number of Patients with Suspected Pneumonia (%) ^1^ N = 200	Number and Prevalence of Confirmed Pneumonia (%)
0	3 (2)	1/3 (33)
1	29 (15)	10/29 (35)
2	76 (38)	49/76 (65)
3	77 (39)	60/77 (78)
4	15 (8)	13/15 (87)

^1^ Percentages not adding to 100 because of rounding.

**Table 4 jcm-08-00514-t004:** Accuracy of physician assessment and score-based algorithm.

	Physician in Charge without LDCT	Score-Based Algorithm (Derivation Cohort)	Score-Based Algorithm (Validation Cohort)	Physician in Charge with LDCT
Proportion of CT scan (%)	0	54	59	100
Sensitivity	95	95	80	92
Specificity	15	48	57	69
Positive predictive value	69	78	66	85
Negative predictive value	59	82	74	81
Positive likelihood ratio	1.1	1.8	2.4	2.9
Negative likelihood ratio	0.4	0.1	0.3	0.1
Diagnostic odd ratio	3	18	8	29
AUROC	0.55 (0.46–0.64)	0.71 (0.63–0.80)	0.69 (0.64–0.74)	0.80 (0.73–0.87)

LDCT: low-dose computed tomography scan. AUROC: area under the receiver operator curve.

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
