# Peer review of "Rational Use of CT-Scan for the Diagnosis of Pneumonia: Comparative Accuracy of Different Strategies"

_jcm, 2019, doi:10.3390/jcm8040514_

Round 1
Reviewer 1 Report
The work has been clearly presented but the proposed CT scan did not give huge advantage over CXR in elderly patients. There were lot of findings that were contradicting the present paradigm in the field assessed in ambulatory patients.
Author Response
Rational use of CT-scan for the diagnosis of pneumonia: comparative accuracy of different strategies
Response to reviews
There follows a point-by-point answer to the reviews. We sincerely thank the three reviewers for their clever and thorough review of our work, which gave us the opportunity to improve greatly the readability of the manuscript.
Reviewer 1
Comments and Suggestions for Authors
The work has been clearly presented but the proposed CT scan did not give huge advantage over CXR in elderly patients. There were lot of findings that were contradicting the present paradigm in the field assessed in ambulatory patients.
Thank you for the revision of our work. We agree that the diagnostic advantage of CT scan is not major. Specifically, the performance of the clinical evaluation supplemented by CT scan in all patients, when compared with expert assessment, is a bit disappointing, as discussed on lines 213-8.
However, we think that the overall clinical impact of CT scan use in the diagnostic approach of pneumonia is currently unknown, and should be the focus of dedicated trials. We added a sentence to discuss this point at the end of the Discussion section:
“Finally, before implementing the proposed score and algorithm in routine use, future studies should focus on validation of its accuracy in larger populations from different settings. As for any diagnostic test, an implementation study should be conducted to test its effectiveness on clinical outcomes compared with the standard diagnostic pathway, and to establish its cost-utility.” (Lines 236-9)
As for ambulatory patients, who are a quite different population with other diagnostic challenges, they were clearly not the focus of the present work.
Reviewer 2 Report
Some comments and question:
Methods:
- Line 76: You mentioned that a low dose CT was performed in all patients after inclusion, and that physician assess the likehood of pneumonia after and before CT, who made the interpretation of CT? was a radiologist?
- Line 87: You considered high and intermediate probabilities as presence of pneumonia what the panel of experters adjudicated. Was the same for the physician in charge? (Line 79)
Results:
- Table 3: the percentages in column 2 do not sum 100%. And in column 3, is not clear to the reader how the percentages are calculated
- Did you have information about respiratory comorbidities of the two cohorts, specially bronchiectasis? because mucus plug can be confused by pulmonary infiltration in the x-ray
- Could you perform an interrater reliability analysis (kappa coefficient) between physician after and before CT with the standard reference?
- In Table 4, you show the accuracy of physician with or without CT in all population, but you could calculate how the accuracy of physician is improved with CT-scan in patients low, intermediate and high probability of pneumonia, or in patient with and algorithm<3points, and="">=3 points. These data would be important for clinical practice.
Discussion is correct. The age difference between the two cohorts is mentioned.
Author Response
Reviewer 2
Comments and Suggestions for Authors
Some comments and question:
Methods:
- Line 76: You mentioned that a low dose CT was performed in all patients after inclusion, and that physician assess the likehood of pneumonia after and before CT, who made the interpretation of CT? was a radiologist?
All CT were interpreted by the radiologist on duty (who was not an expert in thoracic imaging). We clarified this in the method section, lines 76-8
“A LDCT without injection of contrast medium was performed in all patients within 24 hours after inclusion and was interpreted by the radiologist on duty.”
- Line 87: You considered high and intermediate probabilities as presence of pneumonia what the panel of experters adjudicated. Was the same for the physician in charge? (Line 79)
Indeed, we also merged the high and intermediate probabilities for the physician in charge. We clarified this point, line 81
“We merged high and intermediate levels of certainty in one category (presence of pneumonia).”
Results:
- Table 3: the percentages in column 2 do not sum 100%. And in column 3, is not clear to the reader how the percentages are calculated
We added a footnote in table 3
“1Percentages not adding to 100 because of rounding”
We also modified the title of column 3, and added the denominator to clarify the percentage calculated
Number and prevalence of confirmed pneumonia (%) |
1/3 (33) |
10/29 (35) |
49/76 (65) |
60/77 (78) |
13/15 (87) |
- Did you have information about respiratory comorbidities of the two cohorts, specially bronchiectasis? because mucus plug can be confused by pulmonary infiltration in the x-ray
We fully agree with this remark and added the requested information concerning the derivation and validation cohorts
“Forty-seven patients (24%) had a history of chronic respiratory disease, mostly chronic obstructive pulmonary disease, and 16 (8%) had bronchiectasis » (Lines 125-6)
“Chronic respiratory disease was present in 89 patients (28%), and bronchiectasis were identified by LDCT in 52 patients (16%)” (Lines 154-5)
- Could you perform an interrater reliability analysis (kappa coefficient) between physician after and before CT with the standard reference?
Kappa coefficient was 0.12 before, and 0.63 after LDCT. We presented the analysis in the methods section and commented the result as follows:
“Agreement between physician assessment (with and without LDCT) and the reference diagnosis was evaluated with Kappa coefficient.” (Lines 115-7)
“The agreement with the reference diagnosis for physician assessment before LDCT was only slight (kappa coefficient 0.12). After LDCT, physician assessment had substantial agreement with the reference diagnosis (kappa coefficient 0.63). “(Lines 161-3)
- In Table 4, you show the accuracy of physician with or without CT in all population, but you could calculate how the accuracy of physician is improved with CT-scan in patients low, intermediate and high probability of pneumonia, or in patient with and algorithm<3points, and="">=3 points. These data would be important for clinical practice.
Thank you for this remark. We repeated accuracy measurements separately in two strata: < 3 points and >=3 points. Results are displayed thereafter:
Table 4 b. accuracy of physician assessment with and without CT scan in patients with score <=2.< em="">
Physician in charge without LDCT | Physician in charge with LDCT | |
Sensitivity | 93 | 88 |
Specificity | 15 | 67 |
Positive predictive value | 58 | 77 |
Negative predictive value | 64 | 82 |
Positive likelihood ratio | 1.1 | 2.7 |
Negative likelihood ratio | 0.5 | 0.2 |
Diagnostic odd ratio | 2.2 | 13.5 |
AUROC curve | 0.54 (0.43-0.65) | 0.78 (0.68-0.87) |
Table 4 c. accuracy of physician assessment with and without CT scan in patients with score > 2.
Physician in charge without LDCT | Physician in charge with LDCT | |
Sensitivity | 96 | 95 |
Specificity | 16 | 74 |
Positive predictive value | 81 | 93 |
Negative predictive value | 50 | 78 |
Positive likelihood ratio | 1.1 | 3.6 |
Negative likelihood ratio | 0.3 | 0.1 |
Diagnostic odd ratio | 3.7 | 36 |
AUROC curve | 0.56 (0.41-0.71) | 0.84 (0.72-0.96) |
Accuracy of physician assessment without LDCT differs only marginally between the two strata. The AUROC and DOR are higher with LDCT in the > 2 points stratum, but as pre-test probability was very high in these patients (> 80%), we estimated that treatment would be warranted without further testing.
We propose to mention the main results of this stratified analysis without providing the full set of analysis:
“In an analysis stratified on the score, sensitivity and specificity were 93% and 15 % without LDCT in patients with 0 to 2 points, and 88% and 67% after LDCT. Sensitivity and specificity in patients with more than 2 points were 96 % and 16 % before LDCT, and 95% and 74 % after LDCT.” (Lines 165-7)
Discussion is correct. The age difference between the two cohorts is mentioned.

Reviewer 3 Report
A. Brief summary
The authors report the need to build a prediction score, which was can be used to determine indication for performing computed tomography (CT) on pneumonia diagnosis.
The accuracy of two strategies for the diagnosis of pneumonia in elderly patients in the emergency department were compared: one based on a simple prediction model with LDCT performance in patients with predicted low or intermediate probability of disease, and the other based on LDCT performance in all patients.
The authors conclude that a prediction score based on four variables (male gender, acute cough, C-reactive protein and urea) allows superior accuracy for the diagnosis of pneumonia than standard assessment and can guide performance of a CT-scan.
B. Broad comments
1. It is a relevant topic, considering the need of increase of pneumonia diagnostic accuracy, its clinical relevance and the high mortality rates worldwide.
2. The manuscript is well written.
3. The Abstract is concise and adequately summarizes the article content. However, the aim of the study should be clarified:
(Line 27) (…) to build a simple prediction score which was used to determine indication for performing CT-scan.
(Line 63) (…) to compare the accuracy of two strategies for the diagnosis of pneumonia in elderly patients in the emergency department (…)
4. The Introduction represents an adequate synthesis of the literature.
5. “CT” should be primarily identified as “computed tomography” (Line 45, CT-scan)
6. It is suggested the change of “low-dose CT scan (LDCT)” to “low-dose computed tomography (LDCT)”, Line 56 and 57.
7. The manuscript will benefit of a figure with the study design, according with the experimental section described (line 68).
8. According to the authors, the results obtained with the prediction score were validated in a second cohort (319 patients), (lines 102-104). However, it is no clear the results obtained at this validation.
9. There are some errors or uncomplete information at the Tables that should be corrected, namely:
9.1. At the Table 1 (lines 126-128), the P value for the “intermediate and high probability of pneumonia on CXR” is blank. Please complete
9.2. At the Table 1 (lines 126-128), the data for Procalcitonin is 0.36 (0.14-193). Please confirm if the authors want to mention 1.93 instead.
9.3. At the Table 1 (line 126) the “Pneumonia (133) and No pneumonia (67)” should be “Pneumonia (n=133) and No pneumonia (n=67)”.
9.4. At Table 2 (line 133) the “association” should start with capital letter.
9.5. The name of Table 3 (line 139) should be corrected: “Table 3. of patients”.
9.6. It is not clear the mention to “Number of patients” and “Number with pneumonia” at Table 3 (line 139). I believe that the authors intended to mention “Number of patients with suspected pneumonia (n=200)” and “Number of patients with diagnosed pneumonia (n=133)”.
9.7. The “CRP” mentioned at Table 2 was to not previously discriminate. I believe that the authors want to mention “C-reactive protein”.
10. Four characteristics of the patients were independently related with a diagnosis of pneumonia (line 154). Please, identify the four characteristics: male gender, acute cough, C-reactive protein and urea, before described them.
11. There is a very complete description of the study limitations and strengths of the work (lines198-207 and 208-214, respectively).
12. Future studies and perspectives in this thematic should be included.
13. At conclusions (line 218) it is mentioned that “(…) LDCT is indicated for 54% of the patients.” With the tables presented at the manuscript it is not easy to the reader to identify this data.
14. How can other emergency departments implement the prediction score described? It can be clinically implemented using only the four variables identified? It is not clear at the manuscript.
15. The References are not according with the Journal of Clinical Medicine — Instructions for Authors and should be reviewed.
Author Response
Reviewer 3
Comments and Suggestions for Authors
A. Brief summary
The authors report the need to build a prediction score, which was can be used to determine indication for performing computed tomography (CT) on pneumonia diagnosis.
The accuracy of two strategies for the diagnosis of pneumonia in elderly patients in the emergency department were compared: one based on a simple prediction model with LDCT performance in patients with predicted low or intermediate probability of disease, and the other based on LDCT performance in all patients.
The authors conclude that a prediction score based on four variables (male gender, acute cough, C-reactive protein and urea) allows superior accuracy for the diagnosis of pneumonia than standard assessment and can guide performance of a CT-scan.
B. Broad comments
1. It is a relevant topic, considering the need of increase of pneumonia diagnostic accuracy, its clinical relevance and the high mortality rates worldwide.
2. The manuscript is well written.
Thank you very much for your supporting comments
3. The Abstract is concise and adequately summarizes the article content. However, the aim of the study should be clarified:
(Line 27) (…) to build a simple prediction score which was used to determine indication for performing CT-scan.
(Line 63) (…) to compare the accuracy of two strategies for the diagnosis of pneumonia in elderly patients in the emergency department (…)
In accordance with the reviewer comment, we harmonized the description of the aim of the study. The end of the introduction now reads:
“Hence, a prediction model allowing identification of patients whose diagnosis and management is likely to be modified by LDCT would be welcome.
We aimed to build a simple prediction model allowing to determine indication for performing CT-scan, and compared the accuracy of two strategies for the diagnosis of pneumonia in elderly patients in the emergency department: one based on a simple prediction model with LDCT performance in patients with predicted low or intermediate probability of disease, and the other based on LDCT performance in all patients. The reference diagnosis was based on CT-scan imaging and consensus expert evaluation.” (Lines 65-70)
4. The Introduction represents an adequate synthesis of the literature.
5. “CT” should be primarily identified as “computed tomography” (Line 45, CT-scan)
We did the proposed modification
“As a confirmation test, CXR lacks both sensitivity and specificity when compared with computed tomography scan (CT-scan)”
6. It is suggested the change of “low-dose CT scan (LDCT)” to “low-dose computed tomography (LDCT)”, Line 56 and 57.
We corrected the manuscript according to this recommendation
“In a cohort of elderly patients admitted to the hospital for suspected pneumonia, low-dose computed tomography scan (LDCT) modified the probability of pneumonia in 45 % of patients”
7. The manuscript will benefit of a figure with the study design, according with the experimental section described (line 68).
8. According to the authors, the results obtained with the prediction score were validated in a second cohort (319 patients), (lines 102-104). However, it is no clear the results obtained at this validation.
The accuracy of the score in the validation cohort is presented in table 4.
We rephrased the discussion to better present the results obtained in the validation cohort.
“The algorithm had intermediate accuracy (AUROC 0.71, DOR 18). In the validation cohort, the algorithm had similar accuracy, with an AUROC of 0.69 and DOR of 8.” Lines 168-9
9. There are some errors or uncomplete information at the Tables that should be corrected, namely:
9.1. At the Table 1 (lines 126-128), the P value for the “intermediate and high probability of pneumonia on CXR” is blank. Please complete.
Thank you. Indeed, the P value relates to the global difference between the proportions of patient in the three categories. We positioned it one line higher in the table.
Variable | Pneumonia (N=133) | No pneumonia (N=67) | P value |
Demographic characteristics | |||
Age | 83 (78 -89) | 86 (80 -92) | 0.03 |
Male gender | 71 (53) | 28 (42) | 0.18 |
Ambulatory setting (vs nursing home or other) | 117 (88) | 55 (82) | 0.28 |
Symptoms | |||
Acute cough | 120 (90) | 50 (75) | <0.01< span=""> |
Rales | 114 (86) | 57 (85) | 1.00 |
Sputum production | 49 (37) | 25 (37) | 1.00 |
Dyspnea | 95 (71) | 50 (75) | 0.74 |
Chest pain | 26 (20) | 9 (15) | 0.33 |
Confusion | 60 (45) | 32 (48) | 0.60 |
Signs | |||
Heart rate | 90 (78 - 104) | 89 (77 - 101) | 0.53 |
Respiratory rate | 24 (20 -28) | 22 (15 -24) | 0.08 |
Temperature (oC) | 38.0 (37.4 -38.7) | 37.7 (37.1 -38.4) | 0.07 |
Systolic blood pressure | 131 (113 - 148) | 138 (116-156) | 0.20 |
Diastolic blood pressure | 72 (62 -83) | 73 (67 -85) | 0.24 |
Hypoxemia (PaO2<8 kPa or SaO2 < 90%) | 72 (54) | 30 (45) | 0.22 |
Ancillary tests | |||
C-reactive protein (mg/L) | 101 (59 -135) | 63 (38 -108) | <0.01< span=""> |
Leucocytes (G/L) | 12.0 (5.6) | 10.7 (4.0) | 0.09 |
Procalcitonin ug/L | 0.36 (0.14 -1.93) | 0.25 (0.11 -0.66) | 0.04 |
Urea (mmol/L) | 7.7 (5.7-10.8) | 8.3 (6.2-12.9) | 0.04 |
Creatinine (ug/L) | 92 (69-125) | 101 (83-141) | 0.03 |
Probability of pneumonia on CXR (according to physician) Low Intermediate High |
31 (23) 36 (27) 66 (50) |
26 (39) 23 (34) 18 (27) | <0.01< span=""> |
9.2. At the Table 1 (lines 126-128), the data for Procalcitonin is 0.36 (0.14-193). Please confirm if the authors want to mention 1.93 instead.
Thank you for your thorough reading. Indeed, the correct figure was 1.93. We corrected the table accordingly
9.3. At the Table 1 (line 126) the “Pneumonia (133) and No pneumonia (67)” should be “Pneumonia (n=133) and No pneumonia (n=67)”.
Corrected
9.4. At Table 2 (line 133) the “association” should start with capital letter.
Corrected
9.5. The name of Table 3 (line 139) should be corrected: “Table 3. of patients”.
The title now reads “Number of patients with suspected pneumonia and prevalence of confirmed pneumonia according to the score”, see also the following point.
9.6. It is not clear the mention to “Number of patients” and “Number with pneumonia” at Table 3 (line 139). I believe that the authors intended to mention “Number of patients with suspected pneumonia (n=200)” and “Number of patients with diagnosed pneumonia (n=133)”.
Thank you for this opportunity to clarify the table. We renamed the table “Number of patients with suspected pneumonia and prevalence of confirmed pneumonia according to the score”. We also changed the titles of the two columns and the presentation of the results as follows:
Number of points | Number of patients with suspected pneumonia (%)1 N= 200 | Number and prevalence of confirmed pneumonia (%) |
0 | 3 (2) | 1/3 (33) |
1 | 29 (15) | 10/29 (35) |
2 | 76 (38) | 49/76 (65) |
3 | 77 (39) | 60/77 (78) |
4 | 15 (8) | 13/15 (87) |
1Percentages not adding to 100 because of rounding
9.7. The “CRP” mentioned at Table 2 was to not previously discriminate. I believe that the authors want to mention “C-reactive protein”.
Corrected
10. Four characteristics of the patients were independently related with a diagnosis of pneumonia (line 154). Please, identify the four characteristics: male gender, acute cough, C-reactive protein and urea, before described them.
The manuscript now reads: In the multivariate model, four characteristics were independently associated with the diagnosis: male gender, acute cough, C-reactive protein and urea- gender, cough, C-reactive protein and urea (Table 2). (Lines 137-8)
11. There is a very complete description of the study limitations and strengths of the work (lines198-207 and 208-214, respectively).
12. Future studies and perspectives in this thematic should be included.
We fully agree and completed the discussion section with the following sentence:
“Finally, before implementing the proposed score and algorithm in routine use, future studies should focus on validation of its accuracy in larger populations from different settings. As for any diagnostic test, an implementation study should be conducted to test its effectiveness on clinical outcomes compared with the standard diagnostic pathway, and to establish its cost-utility..” (Lines 235-8)
13. At conclusions (line 218) it is mentioned that “(…) LDCT is indicated for 54% of the patients.” With the tables presented at the manuscript it is not easy to the reader to identify this data.
We modified the results section to allow for identification of this data:
The algorithm mandated the performance of a LDCT in 108 patients with less than three points (54% of the population). (Lines 147-8)
14. How can other emergency departments implement the prediction score described? It can be clinically implemented using only the four variables identified? It is not clear at the manuscript.
The diagnostic accuracy of the score should first be validated in larger cohorts from different settings. Moreover, clinical effectiveness and cost-utility studies should be conducted before the use of the score is implemented in clinical practice. See also response to point 12.
15. The References are not according with the Journal of Clinical Medicine — Instructions for Authors and should be reviewed

Round 2
Reviewer 3 Report
The authors have significantly improved the quality of the manuscript.